# CONDITIONAL LEARNING OF FAIR REPRESENTATIONS

**Han Zhao**[*] **& Amanda Coston**
Machine Learning Department
Carnegie Mellon University
`han.zhao@cs.cmu.edu`
`acoston@andrew.cmu.edu`

**Tameem Adel**
Department of Engineering
University of Cambridge
`tah47@cam.ac.uk`

**Geoffrey J. Gordon**
Microsoft Research, Montreal
Machine Learning Department
Carnegie Mellon University
`geoff.gordon@microsoft.com`

## ABSTRACT

We propose a novel algorithm for learning fair representations that can simultaneously mitigate two notions of disparity among different demographic subgroups in the classification setting. Two key components underpinning the design of our algorithm are balanced error rate and conditional alignment of representations. We show how these two components contribute to ensuring accuracy parity and equalized false-positive and false-negative rates across groups without impacting demographic parity. Furthermore, we also demonstrate both in theory and on two real-world experiments that the proposed algorithm leads to a better utility-fairness trade-off on balanced datasets compared with existing algorithms on learning fair representations for classification.

## 1 INTRODUCTION

High-stakes settings, such as loan approvals, criminal justice, and hiring processes, use machine learning tools to help make decisions. A key question in these settings is whether the algorithm makes fair decisions. In settings that have historically had discrimination, we are interested in defining fairness with respect to a protected group, the group which has historically been disadvantaged. The rapidly growing field of algorithmic fairness has a vast literature that proposes various fairness metrics, characterizes the relationship between fairness metrics, and describes methods to build classifiers that satisfy these metrics (Chouldechova & Roth, 2018; Corbett-Davies & Goel, 2018). Among many recent attempts to achieve algorithmic fairness (Dwork et al., 2012; Hardt et al., 2016; Zemel et al., 2013; Zafar et al., 2015), learning fair representations has attracted increasing attention due to its flexibility in learning rich representations based on advances in deep learning (Edwards & Storkey, 2015; Louizos et al., 2015; Beutel et al., 2017; Madras et al., 2018). The backbone idea underpinning this line of work is very intuitive: if the representations of data from different groups are similar to each other, then any classifier acting on such representations will also be agnostic to the group membership.

However, it has long been empirically observed (Calders et al., 2009) and recently been proved (Zhao & Gordon, 2019) that fairness is often at odds with utility. For example, consider demographic parity, which requires the classifier to be independent of the group membership attribute. It is clear that demographic parity will cripple the utility if the demographic group membership and the target variable are indeed correlated. To escape such inherent trade-off, other notions of fairness, such as equalized odds (Hardt et al., 2016), which asks for equal false positive and negative rates across groups, and accuracy parity (Zafar et al., 2017), which seeks equalized error rates across groups, have been proposed. It is a well-known result that equalized odds is incompatible with demographic parity (Barocas et al., 2017) except in degenerate cases where group membership is independent

---

[*]Part of this work was done when Han Zhao was visiting the Vector Institute, Toronto.

of the target variable. Accuracy parity and the so-called predictive rate parity (c.f. Definition 2.4) could be simultaneously achieved, e.g., the COMPAS tool (Dieterich et al., 2016). Furthermore, under some conditions, it is also known that demographic parity can lead to accuracy parity (Zhao & Gordon, 2019). However, whether it is possible to simultaneously guarantee equalized odds and accuracy parity remains an open question.

In this work, we provide an affirmative answer to the above question by proposing an algorithm to align the conditional distributions (on the target variable) of representations across different demographic subgroups. The proposed formulation is a minimax problem that admits a simple reduction to cost-sensitive learning. The key component underpinning the design of our algorithm is the *balanced error rate* (BER, c.f. Section 2) (Feldman et al., 2015; Menon & Williamson, 2018), over the target variable and protected attributes. We demonstrate both in theory and on two real-world experiments that together with the conditional alignment, BER helps our algorithm to simultaneously ensure accuracy parity and equalized odds across groups. Our key contributions are summarized as follows:

- We prove that BER plays a fundamental role in ensuring accuracy parity and a small joint error across groups. Together with the conditional alignment of representations, this implies that we can simultaneously achieve equalized odds and accuracy parity. Furthermore, we also show that when equalized odds is satisfied, BER serves as an upper bound on the error of each subgroup. These results help to justify the design of our algorithm in using BER instead of the marginal error as our loss functions.
- We provide theoretical results that our method achieves equalized odds without impacting demographic parity. This result shows that we can preserve the demographic parity gap for free while simultaneously achieving equalized odds.
- Empirically, among existing fair representation learning methods, we demonstrate that our algorithm is able to achieve a better utility on balanced datasets. On an imbalanced dataset, our algorithm is the only method that achieves accuracy parity; however it does so at the cost of decreased utility.

We believe our theoretical results contribute to the understanding on the relationship between equalized odds and accuracy parity, and the proposed algorithm provides an alternative in real-world scenarios where accuracy parity and equalized odds are desired.

## 2 PRELIMINARY

We first introduce the notations used throughout the paper and formally describe the problem setup and various definitions of fairness explored in the literature.

**Notation** We use $\mathcal{X} \subseteq \mathbb{R}^d$ and $\mathcal{Y} = \{0, 1\}$ to denote the input and output space. Accordingly, we use $X$ and $Y$ to denote the random variables which take values in $\mathcal{X}$ and $\mathcal{Y}$, respectively. Lower case letters $\mathbf{x}$ and $y$ are used to denote the instantiation of $X$ and $Y$. To simplify the presentation, we use $A \in \{0, 1\}$ as the sensitive attribute, e.g., race, gender, etc. [1] Let $\mathcal{H}$ be the hypothesis class of classifiers. In other words, for $h \in \mathcal{H}$, $h : \mathcal{X} \to \mathcal{Y}$ is the predictor that outputs a prediction. Note that even if the predictor does not explicitly take the sensitive attribute $A$ as input, this *fairness through blindness* mechanism can still be biased due to the potential correlations between $X$ and $A$. In this work we study the stochastic setting where there is a joint distribution $\mathcal{D}$ over $X, Y$ and $A$ from which the data are sampled. To keep the notation consistent, for $a, y \in \{0, 1\}$, we use $\mathcal{D}_a$ to mean the conditional distribution of $\mathcal{D}$ given $A = a$ and $\mathcal{D}^y$ to mean the conditional distribution of $\mathcal{D}$ given $Y = y$. For an event $E$, $\mathcal{D}(E)$ denotes the probability of $E$ under $\mathcal{D}$. In particular, in the literature of fair machine learning, we call $\mathcal{D}(Y = 1)$ the *base rate* of distribution $\mathcal{D}$ and we use $\Delta_{\mathrm{BR}}(\mathcal{D}, \mathcal{D}') := |\mathcal{D}(Y = 1) - \mathcal{D}'(Y = 1)|$ to denote the difference of the base rates between two distributions $\mathcal{D}$ and $\mathcal{D}'$ over the same sample space.

Given a feature transformation function $g : \mathcal{X} \to \mathcal{Z}$ that maps instances from the input space $\mathcal{X}$ to feature space $\mathcal{Z}$, we define $g_\sharp \mathcal{D} := \mathcal{D} \circ g^{-1}$ to be the induced (pushforward) distribution of $\mathcal{D}$ under $g$, i.e., for any event $E' \subseteq \mathcal{Z}$, $g_\sharp \mathcal{D}(E') := \mathcal{D}(g^{-1}(E')) = \mathcal{D}(\{x \in \mathcal{X} \mid g(x) \in E'\})$. To measure the discrepancy between distributions, we use $d_{\mathrm{TV}}(\mathcal{D}, \mathcal{D}')$ to denote the total variation between

---

[1] Our main results could also be straightforwardly extended to the setting where $A$ is a categorical variable.

them: $d_{\text{TV}}(\mathcal{D}, \mathcal{D}') := \sup_E |\mathcal{D}(E) - \mathcal{D}'(E)|$. In particular, for binary random variable $Y$, it can be readily verified that the total variation between the marginal distributions $\mathcal{D}(Y)$ and $\mathcal{D}'(Y)$ reduces to the difference of their base rates, $\Delta_{\text{BR}}(\mathcal{D}, \mathcal{D}')$. To see this, realize that $d_{\text{TV}}(\mathcal{D}(Y), \mathcal{D}'(Y)) = \max\{|\mathcal{D}(Y = 1) - \mathcal{D}'(Y = 1)|, |\mathcal{D}(Y = 0) - \mathcal{D}'(Y = 0)|\} = |\mathcal{D}(Y = 1) - \mathcal{D}'(Y = 1)| = \Delta_{\text{BR}}(\mathcal{D}, \mathcal{D}')$ by definition.

Given a joint distribution $\mathcal{D}$, the error of a predictor $h$ under $\mathcal{D}$ is defined as $\text{Err}_{\mathcal{D}}(h) := \mathbb{E}_{\mathcal{D}}[|Y - h(X)|]$. Note that for binary classification problems, when $h(X) \in \{0, 1\}$, $\text{Err}_{\mathcal{D}}(h)$ reduces to the true error rate of binary classification. To make the notation more compact, we may drop the subscript $\mathcal{D}$ when it is clear from the context. Furthermore, we use $\text{CE}_{\mathcal{D}}(\widehat{Y} \,\|\, Y)$ to denote the cross-entropy loss function between the predicted variable $\widehat{Y}$ and the true label distribution $Y$ over the joint distribution $\mathcal{D}$. For binary random variables $Y$, we define $\text{BER}_{\mathcal{D}}(\widehat{Y} \,\|\, Y)$ to be the *balanced error rate* of predicting $Y$ using $\widehat{Y}$, e.g., $\text{BER}_{\mathcal{D}}(\widehat{Y} \,\|\, Y) := \mathcal{D}(\widehat{Y} = 0 \mid Y = 1) + \mathcal{D}(\widehat{Y} = 1 \mid Y = 0)$. Realize that $\mathcal{D}(\widehat{Y} = 0 \mid Y = 1)$ is the false negative rate (FNR) of $\widehat{Y}$ and $\mathcal{D}(\widehat{Y} = 1 \mid Y = 0)$ corresponds to the false positive rate (FPR). So the balanced error rate can also be understood as the sum of FPR and FNR using the predictor $\widehat{Y}$. We can similarly define $\text{BER}_{\mathcal{D}}(\widehat{A} \,\|\, A)$ as well.

**Problem Setup**  In this work we focus on group fairness where the group membership is given by the sensitive attribute $A$. We assume that the sensitive attribute $A$ is available to the learner during training phase, but not inference phase. As a result, post-processing techniques to ensure fairness are not feasible under our setting. In the literature, there are many possible definitions of *fairness* (Narayanan, 2018), and in what follows we provide a brief review of the ones that are mostly relevant to this work.

**Definition 2.1** (Demographic Parity (DP)).  Given a joint distribution $\mathcal{D}$, a classifier $\widehat{Y}$ satisfies *demographic parity* if $\widehat{Y}$ is independent of $A$.

When $\widehat{Y}$ is a deterministic binary classifier, demographic parity reduces to the requirement that $\mathcal{D}_0(\widehat{Y} = 1) = \mathcal{D}_1(\widehat{Y} = 1)$, i.e., positive outcome is given to the two groups at the same rate. Demographic parity is also known as *statistical parity*, and it has been adopted as the default definition of fairness in a series of work (Calders & Verwer, 2010; Calders et al., 2009; Edwards & Storkey, 2015; Johndrow et al., 2019; Kamiran & Calders, 2009; Kamishima et al., 2011; Louizos et al., 2015; Zemel et al., 2013; Madras et al., 2018; Adel et al., 2019). It is not surprising that demographic parity may cripple the utility that we hope to achieve, especially in the common scenario where the *base rates* differ between two groups (Hardt et al., 2016). Formally, the following theorem characterizes the trade-off in terms of the joint error across different groups:

**Theorem 2.1.** (Zhao & Gordon, 2019) Let $\widehat{Y} = h(g(X))$ be the classifier. Then $\text{Err}_{\mathcal{D}_0}(h \circ g) + \text{Err}_{\mathcal{D}_1}(h \circ g) \geq \Delta_{\text{BR}}(\mathcal{D}_0, \mathcal{D}_1) - d_{\text{TV}}(g_\sharp \mathcal{D}_0, g_\sharp \mathcal{D}_1)$.

In this case of representations that are independent of the sensitive attribute $A$, then the second term $d_{\text{TV}}(g_\sharp \mathcal{D}_0, g_\sharp \mathcal{D}_1)$ becomes 0, and this implies:

$$\text{Err}_{\mathcal{D}_0}(h \circ g) + \text{Err}_{\mathcal{D}_1}(h \circ g) \geq \Delta_{\text{BR}}(\mathcal{D}_0, \mathcal{D}_1).$$

At a colloquial level, the above inequality could be understood as an uncertainty principle which says:

> *For fair representations, it is not possible to construct a predictor that simultaneously minimizes the errors on both demographic subgroups.*

More precisely, by the pigeonhole principle, the following corollary holds:

**Corollary 2.1.** If $d_{\text{TV}}(g_\sharp \mathcal{D}_0, g_\sharp \mathcal{D}_1) = 0$, then for any hypothesis $h$, $\max\{\text{Err}_{\mathcal{D}_0}(h \circ g), \text{Err}_{\mathcal{D}_1}(h \circ g)\} \geq \Delta_{\text{BR}}(\mathcal{D}_0, \mathcal{D}_1)/2$.

In words, this means that for fair representations in the demographic parity sense, at least one of the subgroups has to incur a prediction error of at least $\Delta_{\text{BR}}(\mathcal{D}_0, \mathcal{D}_1)/2$ which could be large in settings like criminal justice where $\Delta_{\text{BR}}(\mathcal{D}_0, \mathcal{D}_1)$ is large. In light of such inherent trade-off, an alternative definition is *accuracy parity*, which asks for equalized error rates across different groups:

**Definition 2.2** (Accuracy Parity).  Given a joint distribution $\mathcal{D}$, a classifier $\widehat{Y}$ satisfies *accuracy parity* if $\mathcal{D}_0(\widehat{Y} \neq Y) = \mathcal{D}_1(\widehat{Y} \neq Y)$.

A violation of accuracy parity is also known as disparate mistreatment (Zafar et al., 2017). Different from the definition of demographic parity, the definition of accuracy parity does not eliminate the perfect predictor even when the base rates differ across groups. Of course, other more refined definitions of fairness also exist in the literature, such as *equalized odds* (Hardt et al., 2016).

**Definition 2.3** (Equalized Odds, a.k.a. Positive Rate Parity). Given a joint distribution $\mathcal{D}$, a classifier $\widehat{Y}$ satisfies *equalized odds* if $\widehat{Y}$ is independent of $A$ conditioned on $Y$.

The definition of equalized odds essentially requires equal true positive and false positive rates between different groups, hence it is also known as *positive rate parity*. Analogous to accuracy parity, equalized odds does not eliminate the perfect classifier (Hardt et al., 2016), and we will also justify this observation by formal analysis shortly. Last but not least, we have the following definition for predictive rate parity:

**Definition 2.4** (Predictive Rate Parity). Given a joint distribution $\mathcal{D}$, a classifier $\widehat{Y}$ satisfies *predictive rate parity* if $\mathcal{D}_0(Y = 1 \mid \widehat{Y} = c) = \mathcal{D}_1(Y = 1 \mid \widehat{Y} = c)$, $\forall c \in [0, 1]$.

Note that in the above definition we allow the classifier $\widehat{Y}$ to be probabilistic, meaning that the output of $\widehat{Y}$ could be any value between 0 and 1. For the case where $\widehat{Y}$ is deterministic, Chouldechova (2017) showed that no deterministic classifier can simultaneously satisfy equalized odds and predictive rate parity when the base rates differ across subgroups and the classifier is not perfect.

## 3 Algorithm and Analysis

In this section we first give the proposed optimization formulation and then discuss through formal analysis the motivation of our algorithmic design. Specifically, we show in Section 3.1 why our formulation helps to escape the utility-fairness trade-off. We then in Section 3.2 formally prove that the BERs in the objective function could be used to guarantee a small joint error across different demographic subgroups. In Section 3.3 we establish the relationship between equalized odds and accuracy parity by providing an upper bound of the error gap in terms of both BER and the equalized odds gap. We conclude this section with a brief discussion on the practical implementation of the proposed optimization formulation in Section 3.4. Due to the space limit, we defer all the proofs to the appendix and focus on explaining the high-level intuition and implications of our results.

As briefly discussed in Section 5, a dominant approach in learning fair representations is via adversarial training. Specifically, the following objective function is optimized:

$$\min_{h,g} \max_{h'} \quad \mathrm{CE}_{\mathcal{D}}(h(g(X)) \,\|\, Y) - \lambda \, \mathrm{CE}_{\mathcal{D}}(h'(g(X)) \,\|\, A) \tag{1}$$

In the above formulation, the first term corresponds to minimization of prediction loss of the target variable and the second term represents the loss incurred by the adversary $h'$. Overall this minimax optimization problem expresses a trade-off (controlled by the hyperparameter $\lambda > 0$) between utility and fairness through the representation learning function $g$: on one hand $g$ needs to preserve sufficient information related to $Y$ in order to minimize the first term, but on the other hand $g$ also needs to filter out information related to $A$ in order to maximize the second term.

### 3.1 Conditional Learning of Fair Representations

However, as we introduced in Section 2, the above framework is still subjective to the inherent trade-off between utility and fairness. To escape such a trade-off, we advocate for the following optimization formulation instead:

$$\min_{h,g} \max_{h',h''} \quad \mathrm{BER}_{\mathcal{D}}(h(g(X)) \,\|\, Y) - \lambda \left( \mathrm{BER}_{\mathcal{D}^0}(h'(g(X)) \,\|\, A) + \mathrm{BER}_{\mathcal{D}^1}(h''(g(X)) \,\|\, A) \right) \tag{2}$$

Note that here we optimize over two distinct adversaries, one for each conditional distribution $\mathcal{D}^y$, $y \in \{0, 1\}$. Intuitively, the main difference between (2) and (1) is that we use BER as our objective function in both terms. By definition, since BER corresponds to the sum of Type-I and Type-II errors in classification, the proposed objective function essentially minimizes the conditional errors instead of the original marginal error:

$$\mathcal{D}(\widehat{Y} \neq Y) = \mathcal{D}(Y = 0)\mathcal{D}(\widehat{Y} \neq Y \mid Y = 0) + \mathcal{D}(Y = 1)\mathcal{D}(\widehat{Y} \neq Y \mid Y = 1)$$

$$\mathrm{BER}_{\mathcal{D}}(\widehat{Y} \,\|\, Y) \propto \frac{1}{2}\mathcal{D}(\widehat{Y} \neq Y \mid Y = 0) + \frac{1}{2}\mathcal{D}(\widehat{Y} \neq Y \mid Y = 1), \tag{3}$$

which means that the loss function gives equal importance to the classification error from both groups. Note that the BERs in the second term of (2) are over $\mathcal{D}^y$, $y \in \{0, 1\}$. Roughly speaking, the second term encourages alignment of the the conditional distributions $g_\sharp \mathcal{D}_0^y$ and $g_\sharp \mathcal{D}_1^y$ for $y \in \{0, 1\}$. The following proposition shows that a perfect conditional alignment of the representations also implies that any classifier based on the representations naturally satisfies the equalized odds criterion:

**Proposition 3.1.** For $g : \mathcal{X} \to \mathcal{Z}$, if $d_{\mathrm{TV}}(g_\sharp \mathcal{D}_0^y, g_\sharp \mathcal{D}_1^y) = 0, \forall y \in \{0, 1\}$, then for any classifier $h : \mathcal{Z} \to \{0, 1\}$, $h \circ g$ satisfies equalized odds.

To understand why we aim for conditional alignment of distributions instead of aligning the marginal feature distributions, the following proposition characterizes why such alignment will help us to escape the previous trade-off:

**Proposition 3.2.** For $g : \mathcal{X} \to \mathcal{Z}$, if $d_{\mathrm{TV}}(g_\sharp \mathcal{D}_0^y, g_\sharp \mathcal{D}_1^y) = 0, \forall y \in \{0, 1\}$, then for any classifier $h : \mathcal{Z} \to \{0, 1\}$, $d_{\mathrm{TV}}((h \circ g)_\sharp \mathcal{D}_0, (h \circ g)_\sharp \mathcal{D}_1) \leq \Delta_{\mathrm{BR}}(\mathcal{D}_0, \mathcal{D}_1)$.

As a corollary, this implies that the lower bound given in Theorem 2.1 now vanishes if we instead align the conditional distributions of representations:

$$\mathrm{Err}_{\mathcal{D}_0}(h \circ g) + \mathrm{Err}_{\mathcal{D}_1}(h \circ g) \geq \Delta_{\mathrm{BR}}(\mathcal{D}_0, \mathcal{D}_1) - d_{\mathrm{TV}}((h \circ g)_\sharp \mathcal{D}_0, (h \circ g)_\sharp \mathcal{D}_1) = 0,$$

where the first inequality is due to Lemma 3.1 (Zhao & Gordon, 2019) and the triangle inequality by the $d_{\mathrm{TV}}(\cdot, \cdot)$ distance. Of course, the above lower bound can only serve as a necessary condition but not sufficient to ensure a small joint error across groups. Later (c.f. Theorem 3.2) we will show that together with a small BER on the target variable, it becomes a sufficient condition as well.

## 3.2 THE PRESERVATION OF DEMOGRAPHIC PARITY GAP AND SMALL JOINT ERROR

In this section we show that learning representations by aligning the conditional distributions across groups cannot increase the DP gap as compared to the DP gap of $Y$. Before we proceed, we first introduce a metric to measure the deviation of a predictor from satisfying demographic parity:

**Definition 3.1** (DP Gap). Given a joint distribution $\mathcal{D}$, the *demographic parity gap* of a classifier $\widehat{Y}$ is $\Delta_{\mathrm{DP}}(\widehat{Y}) := |\mathcal{D}_0(\widehat{Y} = 1) - \mathcal{D}_1(\widehat{Y} = 1)|$.

Clearly, if $\Delta_{\mathrm{DP}}(\widehat{Y}) = 0$, then $\widehat{Y}$ satisfies demographic parity. To simplify the exposition, let $\gamma_a := \mathcal{D}_a(Y = 0), \forall a \in \{0, 1\}$. We first prove the following lemma:

**Lemma 3.1.** Assume the conditions in Proposition 3.1 hold and let $\widehat{Y} = h(g(X))$ be the classifier, then $|\mathcal{D}_0(\widehat{Y} = y) - \mathcal{D}_1(\widehat{Y} = y)| \leq |\gamma_0 - \gamma_1| \cdot \left(\mathcal{D}^0(\widehat{Y} = y) + \mathcal{D}^1(\widehat{Y} = y)\right), \forall y \in \{0, 1\}$.

Lemma 3.1 gives an upper bound on the difference of the prediction probabilities across different subgroups. Applying Lemma 3.1 twice for $y = 0$ and $y = 1$, we can prove the following theorem:

**Theorem 3.1.** Assume the conditions in Proposition 3.1 hold and let $\widehat{Y} = h(g(X))$ be the classifier, then $\Delta_{\mathrm{DP}}(\widehat{Y}) \leq \Delta_{\mathrm{BR}}(\mathcal{D}_0, \mathcal{D}_1) = \Delta_{\mathrm{DP}}(Y)$.

**Remark** Theorem 3.1 shows that aligning the conditional distributions of representations between groups will not add more bias in terms of the demographic parity gap. In particular, the DP gap of any classifier that satisfies equalized odds will be at most the DP gap of the perfect classifier. This is particularly interesting as it is well-known in the literature (Barocas et al., 2017) that demographic parity is not compatible with equalized odds except in degenerate cases. Despite this result, Theorem 3.1 says that we can still achieve equalized odds and simultaneously preserve the DP gap.

In Section 3.1 we show that aligning the conditional distributions of representations between groups helps reduce the lower bound of the joint error, but nevertheless that is only a necessary condition. In the next theorem we show that together with a small Type-I and Type-II error in inferring the target variable $Y$, these two properties are also sufficient to ensure a small joint error across different demographic subgroups.

**Theorem 3.2.** Assume the conditions in Proposition 3.1 hold and let $\widehat{Y} = h(g(X))$ be the classifier, then $\mathrm{Err}_{\mathcal{D}_0}(\widehat{Y}) + \mathrm{Err}_{\mathcal{D}_1}(\widehat{Y}) \leq 2\mathrm{BER}_{\mathcal{D}}(\widehat{Y} \,||\, Y)$.

The above bound means that in order to achieve small joint error across groups, it suffices for us to minimize the BER if a classifier satisfies equalized odds. Note that by definition, the BER in the bound equals to the sum of Type-I and Type-II classification errors using $\widehat{Y}$ as a classifier. Theorem 3.2 gives an upper bound of the joint error across groups and it also serves as a motivation for us to design the optimization formulation (2) that simultaneously minimizes the BER and aligns the conditional distributions.

### 3.3 CONDITIONAL ALIGNMENT AND BALANCED ERROR RATES LEAD TO SMALL ERROR

In this section we will see that a small BER and equalized odds together not only serve as a guarantee of a small joint error, but they also lead to a small error gap between different demographic subgroups. Recall that we define $\gamma_a := \mathcal{D}_a(Y = 0), \forall a \in \{0, 1\}$. Before we proceed, we first formally define the accuracy gap and equalized odds gap of a classifier $\widehat{Y}$:

**Definition 3.2** (Error Gap). Given a joint distribution $\mathcal{D}$, the *error gap* of a classifier $\widehat{Y}$ is $\Delta_{\mathrm{Err}}(\widehat{Y}) := |\mathcal{D}_0(\widehat{Y} \neq Y) - \mathcal{D}_1(\widehat{Y} \neq Y)|$.

**Definition 3.3** (Equalized Odds Gap). Given a joint distribution $\mathcal{D}$, the *equalized odds gap* of a classifier $\widehat{Y}$ is $\Delta_{\mathrm{EO}}(\widehat{Y}) := \max_{y \in \{0,1\}} |\mathcal{D}_0^y(\widehat{Y} = 1) - \mathcal{D}_1^y(\widehat{Y} = 1)|$.

By definition the error gap could also be understood as the accuracy parity gap between different subgroups. The following theorem characterizes the relationship between error gap, equalized odds gap and the difference of base rates across subgroups:

**Theorem 3.3.** For any classifier $\widehat{Y}$, $\Delta_{\mathrm{Err}}(\widehat{Y}) \leq \Delta_{\mathrm{BR}}(\mathcal{D}_0, \mathcal{D}_1) \cdot \mathrm{BER}_{\mathcal{D}}(\widehat{Y} \parallel Y) + 2\Delta_{\mathrm{EO}}(\widehat{Y})$.

As a direct corollary of Theorem 3.3, if the classifier $\widehat{Y}$ satisfies equalized odds, then $\Delta_{\mathrm{EO}}(\widehat{Y}) = 0$. In this case since $\Delta_{\mathrm{BR}}(\mathcal{D}_0, \mathcal{D}_1)$ is a constant, minimizing the balanced error rate $\mathrm{BER}_{\mathcal{D}}(\widehat{Y} \parallel Y)$ also leads to minimizing the error gap. Furthermore, if we combine Theorem 3.2 and Theorem 3.3 together, we can guarantee that each of the errors cannot be too large:

**Corollary 3.1.** For any joint distribution $\mathcal{D}$ and classifier $\widehat{Y}$, if $\widehat{Y}$ satisfies equalized odds, then $\max\{\mathrm{Err}_{\mathcal{D}_0}(\widehat{Y}), \mathrm{Err}_{\mathcal{D}_1}(\widehat{Y})\} \leq \Delta_{\mathrm{BR}}(\mathcal{D}_0, \mathcal{D}_1) \cdot \mathrm{BER}_{\mathcal{D}}(\widehat{Y} \parallel Y)/2 + \mathrm{BER}_{\mathcal{D}}(\widehat{Y} \parallel Y)$.

**Remark** It is a well-known fact that out of the three fairness criteria, i.e., demographic parity, equalized odds, and predictive rate parity, any two of them cannot hold simultaneously (Barocas et al., 2017) except in degenerate cases. By contrast, Theorem 3.3 suggests it *is* possible to achieve both equalized odds and accuracy parity. In particular, among all classifiers that satisfy equalize odds, it suffices to minimize the sum of Type-I and Type-II error $\mathrm{BER}_{\mathcal{D}}(\widehat{Y} \parallel Y)$ in order to achieve accuracy parity. It is also worth pointing out that Theorem 3.3 provides only an upper bound, but not necessarily the tightest one. In particular, the error gap could still be 0 while $\mathrm{BER}_{\mathcal{D}}(\widehat{Y} \parallel Y)$ is greater than 0. To see this, we have

$$\begin{cases} \mathrm{Err}_{\mathcal{D}_0}(\widehat{Y}) = \mathcal{D}_0(Y = 0) \cdot \mathcal{D}_0(\widehat{Y} = 1 \mid Y = 0) + \mathcal{D}_0(Y = 1) \cdot \mathcal{D}_0(\widehat{Y} = 0 \mid Y = 1) \\ \mathrm{Err}_{\mathcal{D}_1}(\widehat{Y}) = \mathcal{D}_1(Y = 0) \cdot \mathcal{D}_1(\widehat{Y} = 1 \mid Y = 0) + \mathcal{D}_1(Y = 1) \cdot \mathcal{D}_1(\widehat{Y} = 0 \mid Y = 1). \end{cases}$$

Now if the predictor $\widehat{Y}$ satisfies equalized odds, then

$$\mathcal{D}_0(\widehat{Y} = 1 \mid Y = 0) = \mathcal{D}_1(\widehat{Y} = 1 \mid Y = 0) = \mathcal{D}(\widehat{Y} = 1 \mid Y = 0),$$

$$\mathcal{D}_0(\widehat{Y} = 0 \mid Y = 1) = \mathcal{D}_1(\widehat{Y} = 0 \mid Y = 1) = \mathcal{D}(\widehat{Y} = 0 \mid Y = 1).$$

Hence the error gap $\Delta_{\mathrm{Err}}(\widehat{Y})$ admits the following identity:

$$\Delta_{\mathrm{Err}}(\widehat{Y}) = \left| \mathcal{D}(\widehat{Y} = 1 \mid Y = 0)\big(\mathcal{D}_0(Y = 0) - \mathcal{D}_1(Y = 0)\big) + \mathcal{D}(\widehat{Y} = 0 \mid Y = 1)\big(\mathcal{D}_0(Y = 1) - \mathcal{D}_1(Y = 1)\big) \right|$$

$$= \Delta_{\mathrm{BR}}(\mathcal{D}_0, \mathcal{D}_1) \cdot \left| \mathcal{D}(\widehat{Y} = 1 \mid Y = 0) - \mathcal{D}(\widehat{Y} = 0 \mid Y = 1) \right|$$

$$= \Delta_{\mathrm{BR}}(\mathcal{D}_0, \mathcal{D}_1) \cdot \left| \mathrm{FPR}(\widehat{Y}) - \mathrm{FNR}(\widehat{Y}) \right|.$$

In other words, if the predictor $\widehat{Y}$ satisfies equalized odds, then in order to have equalized accuracy, $\widehat{Y}$ only needs to have equalized FPR and FNR *globally* when the base rates differ across groups. This is a much weaker condition to ask for than the one asking $\mathrm{BER}_{\mathcal{D}}(\widehat{Y} \parallel Y) = 0$.

## 3.4 PRACTICAL IMPLEMENTATION

We cannot directly optimize the proposed optimization formulation (2) since the binary $0/1$ loss is NP-hard to optimize, or even approximately optimize over a wide range of hypothesis classes (Ben-David et al., 2003). However, observe that for any classifier $\widehat{Y}$ and $y \in \{0, 1\}$, the log-loss (cross-entropy loss) $\mathrm{CE}_{\mathcal{D}^y}(\widehat{Y} \| Y)$ is a convex relaxation of the binary loss:

$$\mathcal{D}(\widehat{Y} \neq y \mid Y = y) = \frac{\mathcal{D}(\widehat{Y} \neq y, Y = y)}{\mathcal{D}(Y = y)} \leq \frac{\mathrm{CE}_{\mathcal{D}^y}(\widehat{Y} \| Y)}{\mathcal{D}(Y = y)}. \tag{4}$$

Hence in practice we can relax the optimization problem (2) to a cost-sensitive cross-entropy loss minimization problem, where the weight for each class is given by the inverse marginal probability of the corresponding class. This allows us to equivalently optimize the objective function without explicitly computing the conditional distributions.

## 4 EMPIRICAL STUDIES

In light of our theoretic findings, in this section we verify the effectiveness of the proposed algorithm in simultaneously ensuring equalized odds and accuracy parity using real-world datasets. We also analyze the impact of imposing such parity constraints on the utility of the target classifier, as well as its relationship to the intrinsic structure of the binary classification problem, e.g., the difference of base rates across groups, the global imbalance of the target variable, etc. We analyze how this imbalance affects the utility-fairness trade-off. As we shall see shortly, we will empirically demonstrate that, in many cases, especially the ones where the dataset is imbalanced in terms of the target variable, this will inevitably compromise the target utility. While for balanced datasets, this trend is less obvious: the proposed algorithm achieves a better fairness-utility trade-off when compared with existing fair representation learning methods and we can hope to achieve fairness without sacrificing too much on utility.

Table 1: Statistics about the Adult and COMPAS datasets.

|  | Train / Test | $\mathcal{D}_0(Y = 1)$ | $\mathcal{D}_1(Y = 1)$ | $\Delta_{\mathrm{BR}}(\mathcal{D}_0, \mathcal{D}_1)$ | $\mathcal{D}(Y = 1)$ | $\mathcal{D}(A = 1)$ |
|---|---|---|---|---|---|---|
| **Adult** | $30,162/15,060$ | $0.310$ | $0.113$ | $0.196$ | $0.246$ | $0.673$ |
| **COMPAS** | $4,320/1,852$ | $0.400$ | $0.529$ | $0.129$ | $0.467$ | $0.514$ |

## 4.1 EXPERIMENTAL SETUP

To this end, we perform experiments on two popular real-world datasets in the literature of algorithmic fairness, including an income-prediction dataset, known as the *Adult* dataset, from the UCI Machine Learning Repository (Dua & Graff, 2017), and the Propublica *COMPAS* dataset (Dieterich et al., 2016). The basic statistics of these two datasets are listed in Table 1.

**Adult**  Each instance in the Adult dataset describes an adult, e.g., gender, education level, age, etc, from the 1994 US Census. In this dataset we use gender as the sensitive attribute, and the processed data contains 114 attributes. The target variable (income) is also binary: 1 if $\geq$ 50K/year otherwise 0. For the sensitive attribute $A$, $A = 0$ means male otherwise female. From Table 1 we can see that the base rates are quite different ($0.310$ vs. $0.113$) across groups in the Adult dataset. The dataset is also imbalanced in the sense that only around $24.6\%$ of the instances have target label 1. Furthermore, the group ratio is also imbalanced: roughly $67.3\%$ of the data are male.

**COMPAS**  The goal of the COMPAS dataset is binary classification on whether a criminal defendant will recidivate within two years or not. Each instance contains 12 attributes, including age, race, gender, number of prior crimes, etc. For this dataset, we use the race (white $A = 0$ vs. black $A = 1$) as our sensitive attribute and target variable is 1 iff recidivism. The base rates are different across groups, but the COMPAS dataset is balanced in both the target variable and the sensitive attribute.

To validate the effect of ensuring equalized odds and accuracy parity, for each dataset, we perform controlled experiments by fixing the baseline network architecture so that it is shared among all the fair representation learning methods. We term the proposed method CFAIR (for conditional fair representations) that minimizes conditional errors both the target variable loss function and adversary loss function. To demonstrate the importance of using BER in the loss function of target variable, we compare with a variant of CFAIR that only uses BER in the loss of adversaries, denoted as CFAIR-EO. To see the relative effect of using cross-entropy loss vs $L_1$ loss, we also show the results of LAFTR (Madras et al., 2018), a state-of-the-art method for learning fair representations. Note that LAFTR is closely related to CFAIR-EO but slightly different: LAFTR uses global cross-entropy loss for target variable, but conditional $L_1$ loss for the adversary. Also, there is only one adversary in LAFTR, while there are two adversaries, one for $\mathcal{D}^0$ and one for $\mathcal{D}^1$, in both CFAIR and CFAIR-EO. Lastly, we also present baseline results of FAIR (Edwards & Storkey, 2015), which aims for demographic parity representations, and NODEBIAS, the baseline network without any fairness constraint. For all the fair representation learning methods, we use the gradient reversal layer (Ganin et al., 2016) to implement the gradient descent ascent (GDA) algorithm to optimize the minimax problem. All the experimental details, including network architectures, learning rates, batch sizes, etc. are provided in the appendix.

## 4.2 RESULTS AND ANALYSIS

In Figure 1 and Figure 2 we show the error gap $\Delta_{\mathrm{Err}}$, equalized odds gap $\Delta_{\mathrm{EO}}$, demographic parity gap $\Delta_{\mathrm{DP}}$ and the joint error across groups $\mathrm{Err}_0 + \mathrm{Err}_1$ of the aforementioned fair representation learning algorithms on both the Adult and the COMPAS datasets. For each algorithm and dataset, we also gradually increase the value of the trade-off parameter $\lambda$ and compute the corresponding metrics.

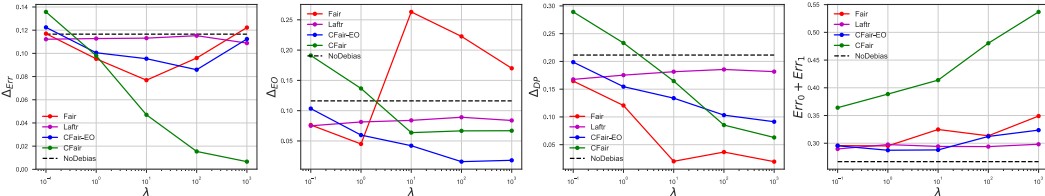

Figure 1: The error gap $\Delta_{\mathrm{Err}}$, equalized odds gap $\Delta_{\mathrm{EO}}$, demographic parity gap $\Delta_{\mathrm{DP}}$ and joint error $\mathrm{Err}_0 + \mathrm{Err}_1$ on the Adult dataset with $\lambda \in \{0.1, 1.0, 10.0, 100.0, 1000.0\}$.

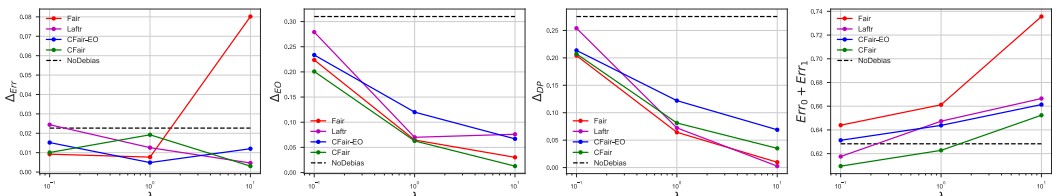

Figure 2: The error gap $\Delta_{\mathrm{Err}}$, equalized odds gap $\Delta_{\mathrm{EO}}$, demographic parity gap $\Delta_{\mathrm{DP}}$ and joint error $\mathrm{Err}_0 + \mathrm{Err}_1$ on the COMPAS dataset with $\lambda \in \{0.1, 1.0, 10.0\}$.

**Adult**  Due to the imbalance of $A$ in the Adult dataset, in the first plot of Figure 1 we can see that all the algorithms except CFAIR have a large error gap of around 0.12. As a comparison, observe that the error gap of CFAIR when $\lambda = 1e3$ almost reduces to 0, confirming the effectiveness of our algorithm in ensuring accuracy parity. From the second plot, we can verify that all the three methods, including CFAIR, CFAIR-EO and LAFTR successfully ensure a small equalized odds gap, and they also decrease demographic parity gaps (the third plot). FAIR is the most effective one in mitigating $\Delta_{\mathrm{DP}}$ since its objective function directly optimizes for that goal. Note that from the second plot we can also confirm that CFAIR-EO is more effective than LAFTR in reducing $\Delta_{\mathrm{EO}}$. The reason is two-fold. First, CFAIR-EO uses two distinct adversaries and hence it effectively competes with

stronger adversaries than LAFTR. Second, CFAIR-EO uses the cross-entropy loss instead of the $L_1$ loss for the adversary, and it is well-known that the maximum-likelihood estimator (equivalent to using the cross-entropy loss) is asymptotically efficient and optimal. On the other hand, since the Adult dataset is imbalanced (in terms of $Y$), using BER in the loss function of the target variable can thus to a large extent hurt the utility, and this is also confirmed from the last plot, where we show the joint error.

**COMPAS** The first three plots of Figure 2 once again verify that CFAIR successfully leads to reduced error gap, equalized odds gap and also demographic parity gap. These experimental results are consistent with our theoretical findings where we show that if the representations satisfy equalized odds, then its $\Delta_{DP}$ cannot exceed that of the optimal classifier, as shown by the horizontal dashed line in the third plot. In the fourth plot of Figure 2, we can see that as we increase $\lambda$, all the fair representation learning algorithms sacrifice utility. However, in contrast to Figure 1, here the proposed algorithm CFAIR has the smallest trade-off: this shows that CFAIR is particularly suited in the cases when the dataset is balanced and we would like to simultaneously ensure accuracy parity and equalized odds. As a comparison, while CFAIR-EO is still effective, it is slightly worse than CFAIR in terms of both ensuring parity and achieving small joint error.

## 5 RELATED WORK

**Algorithmic Fairness** In the literature of algorithmic fairness, two key notions of fairness have been extensively proposed and explored, i.e., *group fairness*, including various variants defined in Section 2, and *individual fairness*, which means that similar individuals should be treated similarly. Due to the complexity in defining a distance metric to measure the similarity between individuals (Dwork et al., 2012), most recent research focuses on designing efficient algorithms to achieve group fairness (Zemel et al., 2013; Zafar et al., 2015; Hardt et al., 2016; Zafar et al., 2017; Madras et al., 2018; Creager et al., 2019; Madras et al., 2019). In particular, Hardt et al. (2016) proposed a post-processing technique to achieve equalized odds by taking as input the model's prediction and the sensitive attribute. However, the post-processing technique requires access to the sensitive attribute during the inference phase, which is often not available in many real-world scenarios. Another line of work uses causal inference to define notions of causal fairness and to formulate procedures for achieving these notions (Zhang et al., 2018; Wang et al., 2019; Madras et al., 2019; Kilbertus et al., 2017; Kusner et al., 2017; Nabi & Shpitser, 2018). These approaches require making untestable assumptions. Of particular note is the observation in Coston et al. (2019) that fairness-adjustment procedures based on $Y$ in settings with treatment effects may lead to adverse outcomes. To apply our method in such settings, we would need to match conditional counterfactual distributions, which could be a direction of future research.

**Theoretical Results on Fairness** Theoretical work studying the relationship between different kinds of fairness notions are abundant. Motivated by the controversy of the potential discriminatory bias in recidivism prediction instruments, Chouldechova (2017) showed an intrinsic incompatibility between equalized odds and predictive rate parity. In the seminal work of Kleinberg et al. (2016), the authors demonstrated that when the base rates differ between different groups, then a non-perfect classifier cannot simultaneously be statistically calibrated and satisfy equalized odds. In the context of cost-sensitive learning, Menon & Williamson (2018) show that if the optimal decision function is dissimilar to a fair decision, then the fairness constraint will not significantly harm the target utility. The idea of reducing fair classification to cost-sensitive learning is not new. Agarwal et al. (2018) explored the connection between fair classification and a sequence of cost-sensitive learning problems where each stage corresponds to solving a linear minimax saddle point problem. In a recent work (Zhao & Gordon, 2019), the authors proved a lower bound on the joint error across different groups when a classifier satisfies demographic parity. They also showed that when the decision functions are close between groups, demographic parity also implies accuracy parity. The theoretical results in this work establish a relationship between accuracy parity and equalized odds: these two fairness notions are fundamentally related by the base rate gap and the balanced error rate. Furthermore, we also show that for any predictor that satisfies equalized odds, the balanced error rate also serves as an upper bound on the joint error across demographic subgroups.

**Fair Representations** Through the lens of representation learning, recent advances in building fair algorithmic decision making systems focus on using adversarial methods to learn fair representations

that also preserve sufficient information for the prediction task (Edwards & Storkey, 2015; Beutel et al., 2017; Zhang et al., 2018; Madras et al., 2018; Adel et al., 2019). In a nutshell, the key idea is to frame the problem of learning fair representations as a two-player game, where the data owner is competing against an adversary. The goal of the adversary is to infer the group attribute as much as possible while the goal of the data owner is to remove information related to the group attribute and simultaneously to preserve utility-related information for accurate prediction. Apart from using adversarial classifiers to enforce group fairness, other distance metrics have also been used to learn fair representations, e.g., the maximum mean discrepancy (Louizos et al., 2015), and the Wasserstein-1 distance (Jiang et al., 2019). In contrast to these methods, in this paper we advocate for optimizing BER on both the target loss and adversary loss in order to simultaneously achieve accuracy parity and equalized odds. We also show that this leads to better utility-fairness trade-off for balanced datasets.

## 6 Conclusion and Future Work

In this paper we propose a novel representation learning algorithm that aims to simultaneously ensure accuracy parity and equalized odds. The main idea underlying the design of our algorithm is to align the conditional distributions of representations (rather than marginal distributions) and use balanced error rate (i.e., the conditional error) on both the target variable and the sensitive attribute. Theoretically, we prove how these two concepts together help to ensure accuracy parity and equalized odds without impacting demographic parity, and we also show how these two can be used to give a guarantee on the joint error across different demographic subgroups. Empirically, we demonstrate on two real-world experiments that the proposed algorithm effectively leads to the desired notions of fairness, and it also leads to better utility-fairness trade-off on balanced datasets.

**Calibration and Utility**   Our work takes a step towards better understanding the relationships between different notions of fairness and their corresponding trade-off with utility. In some scenarios, e.g., the COMPAS tool, it is desired to have a decision making system that is also well calibrated. While it is well-known that statistical calibration is not compatible with demographic parity or equalized odds, from a theoretical standpoint it is still not clear whether calibration will harm utility and if so, what is the fundamental limit of a calibrated tool on utility.

**Fairness and Privacy**   Future work could also investigate how to make use of the close relationship between privacy and group fairness. At a colloquial level, fairness constraints require a predictor to be (to some extent) agnostic about the group membership attribute. The membership query attack in privacy asks the same question – is it possible to guarantee that even an optimal adversary cannot steal personal information through inference attacks. Prior work (Dwork et al., 2012) has described the connection between the notion of individual fairness and differential privacy. Hence it would be interesting to exploit techniques developed in the literature of privacy to develop more efficient fairness-aware learning algorithms. On the other hand, results obtained in the algorithmic fairness literature could also potentially lead to better privacy-preserving machine learning algorithms (Zhao et al., 2019).

### Acknowledgments

HZ and GG would like to acknowledge support from the DARPA XAI project, contract #FA87501720152 and an Nvidia GPU grant. HZ would also like to thank Richard Zemel, Toniann Pitassi, David Madras and Elliot Creager for helpful discussions during HZ's visit to the Vector Institute.

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

## A PROOFS

### A.1 PROOF OF PROPOSITION 3.1

**Proposition 3.1.** For $g : \mathcal{X} \to \mathcal{Z}$, if $d_{\text{TV}}(g_\sharp \mathcal{D}_0^y, g_\sharp \mathcal{D}_1^y) = 0, \forall y \in \{0, 1\}$, then for any classifier $h : \mathcal{Z} \to \{0, 1\}$, $h \circ g$ satisfies equalized odds.

*Proof.* To prove this proposition, we first show that for any pair of distributions $\mathcal{D}, \mathcal{D}'$ over $\mathcal{Z}$ and any hypothesis $h : \mathcal{Z} \to \{0, 1\}$, $d_{\text{TV}}(h_\sharp \mathcal{D}, h_\sharp \mathcal{D}') \le d_{\text{TV}}(\mathcal{D}, \mathcal{D}')$. Note that since $h$ is a hypothesis, there are only two events in the induced probability space, i.e., $h(\cdot) = 0$ or $h(\cdot) = 1$. Hence by definition of the induced (pushforward) distribution, we have:

$$d_{\text{TV}}(h_\sharp \mathcal{D}, h_\sharp \mathcal{D}') = \max_{E = h^{-1}(0), \text{ or } E = h^{-1}(1)} |\mathcal{D}(E) - \mathcal{D}'(E)|$$
$$\le \sup_{E \text{ is measurable subset of } \mathcal{Z}} |\mathcal{D}(E) - \mathcal{D}'(E)|$$
$$= d_{\text{TV}}(\mathcal{D}, \mathcal{D}').$$

Apply the above inequality twice for $y \in \{0, 1\}$:

$$0 \le d_{\text{TV}}((h \circ g)_\sharp \mathcal{D}_0^y, (h \circ g)_\sharp \mathcal{D}_1^y) \le d_{\text{TV}}(g_\sharp \mathcal{D}_0^y, g_\sharp \mathcal{D}_1^y) = 0,$$

meaning

$$d_{\text{TV}}((h \circ g)_\sharp \mathcal{D}_0^y, (h \circ g)_\sharp \mathcal{D}_1^y) = 0,$$

which further implies that $h(g(X))$ is independent of $A$ given $Y = y$ since $h(g(X))$ is binary. ∎

### A.2 PROOF OF PROPOSITION 3.2

**Proposition 3.2.** For $g : \mathcal{X} \to \mathcal{Z}$, if $d_{\text{TV}}(g_\sharp \mathcal{D}_0^y, g_\sharp \mathcal{D}_1^y) = 0, \forall y \in \{0, 1\}$, then for any classifier $h : \mathcal{Z} \to \{0, 1\}$, $d_{\text{TV}}((h \circ g)_\sharp \mathcal{D}_0, (h \circ g)_\sharp \mathcal{D}_1) \le \Delta_{\text{BR}}(\mathcal{D}_0, \mathcal{D}_1)$.

*Proof.* Let $\widehat{Y} = (h \circ g)(X)$ and note that $\widehat{Y}$ is binary, we have

$$d_{\text{TV}}((h \circ g)_\sharp \mathcal{D}_0, (h \circ g)_\sharp \mathcal{D}_1) = \frac{1}{2} \left( |\mathcal{D}_0(\widehat{Y} = 0) - \mathcal{D}_1(\widehat{Y} = 0)| + |\mathcal{D}_0(\widehat{Y} = 1) - \mathcal{D}_1(\widehat{Y} = 1)| \right).$$

Now, by Proposition 3.1, if $d_{\text{TV}}(g_\sharp \mathcal{D}_0^y, g_\sharp \mathcal{D}_1^y) = 0, \forall y \in \{0, 1\}$, it follows that $d_{\text{TV}}((h \circ g)_\sharp \mathcal{D}_0^y, (h \circ g)_\sharp \mathcal{D}_1^y) = 0, \forall y \in \{0, 1\}$ as well. Applying Lemma 3.1, we know that $\forall y \in \{0, 1\}$,

$$|\mathcal{D}_0(\widehat{Y} = y) - \mathcal{D}_1(\widehat{Y} = y)| \le |\mathcal{D}_0(Y = 0) - \mathcal{D}_1(Y = 0)| \cdot \left( \mathcal{D}^0(\widehat{Y} = y) + \mathcal{D}^1(\widehat{Y} = y) \right).$$

Hence,

$$d_{\text{TV}}((h \circ g)_\sharp \mathcal{D}_0, (h \circ g)_\sharp \mathcal{D}_1) = \frac{1}{2} \left( |\mathcal{D}_0(\widehat{Y} = 0) - \mathcal{D}_1(\widehat{Y} = 0)| + |\mathcal{D}_0(\widehat{Y} = 1) - \mathcal{D}_1(\widehat{Y} = 1)| \right)$$
$$\le \frac{|\mathcal{D}_0(Y = 0) - \mathcal{D}_1(Y = 0)|}{2} \left( \left( \mathcal{D}^0(\widehat{Y} = 0) + \mathcal{D}^1(\widehat{Y} = 0) \right) + \left( \mathcal{D}^0(\widehat{Y} = 1) + \mathcal{D}^1(\widehat{Y} = 1) \right) \right)$$
$$= \frac{|\mathcal{D}_0(Y = 0) - \mathcal{D}_1(Y = 0)|}{2} \left( \left( \mathcal{D}^0(\widehat{Y} = 0) + \mathcal{D}^0(\widehat{Y} = 1) \right) + \left( \mathcal{D}^1(\widehat{Y} = 0) + \mathcal{D}^1(\widehat{Y} = 1) \right) \right)$$
$$= \frac{|\mathcal{D}_0(Y = 0) - \mathcal{D}_1(Y = 0)|}{2} \cdot 2$$
$$= |\mathcal{D}_0(Y = 0) - \mathcal{D}_1(Y = 0)|$$
$$= \Delta_{\text{BR}}(\mathcal{D}_0, \mathcal{D}_1). \qquad \blacksquare$$

### A.3 PROOF OF LEMMA 3.1

Recall that we define $\gamma_a := \mathcal{D}_a(Y = 0), \forall a \in \{0, 1\}$.

**Lemma 3.1.** Assume the conditions in Proposition 3.1 hold and let $\widehat{Y} = h(g(X))$ be the classifier, then $|\mathcal{D}_0(\widehat{Y} = y) - \mathcal{D}_1(\widehat{Y} = y)| \le |\gamma_0 - \gamma_1| \cdot \left( \mathcal{D}^0(\widehat{Y} = y) + \mathcal{D}^1(\widehat{Y} = y) \right), \forall y \in \{0, 1\}$.

*Proof.* To bound $|\mathcal{D}_0(\widehat{Y} = y) - \mathcal{D}_1(\widehat{Y} = y)|$, for $y \in \{0, 1\}$, by the law of total probability, we have:

$$|\mathcal{D}_0(\widehat{Y} = y) - \mathcal{D}_1(\widehat{Y} = y)| =$$
$$= \left|\left(\mathcal{D}_0^0(\widehat{Y} = y)\mathcal{D}_0(Y = 0) + \mathcal{D}_0^1(\widehat{Y} = y)\mathcal{D}_0(Y = 1)\right) - \left(\mathcal{D}_1^0(\widehat{Y} = y)\mathcal{D}_1(Y = 0) + \mathcal{D}_1^1(\widehat{Y} = y)\mathcal{D}_1(Y = 1)\right)\right|$$
$$\leq \left|\gamma_0 \mathcal{D}_0^0(\widehat{Y} = y) - \gamma_1 \mathcal{D}_1^0(\widehat{Y} = y)\right| + \left|(1 - \gamma_0)\mathcal{D}_0^1(\widehat{Y} = y) - (1 - \gamma_1)\mathcal{D}_1^1(\widehat{Y} = y)\right|,$$

where the above inequality is due to the triangular inequality. Now apply Proposition 3.1, we know that $\widehat{Y}$ satisfies equalized odds, so we have $\mathcal{D}_0^0(\widehat{Y} = y) = \mathcal{D}_1^0(\widehat{Y} = y) = \mathcal{D}^0(\widehat{Y} = y)$ and $\mathcal{D}_0^1(\widehat{Y} = y) = \mathcal{D}_1^1(\widehat{Y} = y) = \mathcal{D}^1(\widehat{Y} = y)$, leading to:

$$= \left|\gamma_0 - \gamma_1\right| \cdot \mathcal{D}^0(\widehat{Y} = y) + \left|(1 - \gamma_0) - (1 - \gamma_1)\right| \cdot \mathcal{D}^1(\widehat{Y} = y)$$
$$= \left|\gamma_0 - \gamma_1\right| \cdot \left(\mathcal{D}^0(\widehat{Y} = y) + \mathcal{D}^1(\widehat{Y} = y)\right),$$

which completes the proof. ∎

## A.4  PROOF OF THEOREM 3.1

**Theorem 3.1.** Assume the conditions in Proposition 3.1 hold and let $\widehat{Y} = h(g(X))$ be the classifier, then $\Delta_{\mathrm{DP}}(\widehat{Y}) \leq \Delta_{\mathrm{BR}}(\mathcal{D}_0, \mathcal{D}_1) = \Delta_{\mathrm{DP}}(Y)$.

*Proof.* To bound $\Delta_{\mathrm{DP}}(\widehat{Y})$, realize that $|\mathcal{D}_0(\widehat{Y} = 0) - \mathcal{D}_1(\widehat{Y} = 0)| = |\mathcal{D}_0(\widehat{Y} = 1) - \mathcal{D}_1(\widehat{Y} = 1)|$, so we can rewrite the DP gap as follows:

$$\Delta_{\mathrm{DP}}(\widehat{Y}) = \frac{1}{2}\left(|\mathcal{D}_0(\widehat{Y} = 0) - \mathcal{D}_1(\widehat{Y} = 0)| + |\mathcal{D}_0(\widehat{Y} = 1) - \mathcal{D}_1(\widehat{Y} = 1)|\right).$$

Now apply Lemma 3.1 twice for $y = 0$ and $y = 1$, we have:

$$|\mathcal{D}_0(\widehat{Y} = 0) - \mathcal{D}_1(\widehat{Y} = 0)| \leq |\gamma_0 - \gamma_1| \cdot \left(\mathcal{D}^0(\widehat{Y} = 0) + \mathcal{D}^1(\widehat{Y} = 0)\right)$$
$$|\mathcal{D}_0(\widehat{Y} = 1) - \mathcal{D}_1(\widehat{Y} = 1)| \leq |\gamma_0 - \gamma_1| \cdot \left(\mathcal{D}^0(\widehat{Y} = 1) + \mathcal{D}^1(\widehat{Y} = 1)\right).$$

Taking sum of the above two inequalities yields

$$|\mathcal{D}_0(\widehat{Y} = 0) - \mathcal{D}_1(\widehat{Y} = 0)| + |\mathcal{D}_0(\widehat{Y} = 1) - \mathcal{D}_1(\widehat{Y} = 1)|$$
$$\leq |\gamma_0 - \gamma_1|\left(\left(\mathcal{D}^0(\widehat{Y} = 0) + \mathcal{D}^1(\widehat{Y} = 0)\right) + \left(\mathcal{D}^0(\widehat{Y} = 1) + \mathcal{D}^1(\widehat{Y} = 1)\right)\right)$$
$$= |\gamma_0 - \gamma_1|\left(\left(\mathcal{D}^0(\widehat{Y} = 0) + \mathcal{D}^0(\widehat{Y} = 1)\right) + \left(\mathcal{D}^1(\widehat{Y} = 0) + \mathcal{D}^1(\widehat{Y} = 1)\right)\right)$$
$$= 2|\gamma_0 - \gamma_1|.$$

Combining all the inequalities above, we know that

$$\Delta_{\mathrm{DP}}(\widehat{Y}) = \frac{1}{2}\left(|\mathcal{D}_0(\widehat{Y} = 0) - \mathcal{D}_1(\widehat{Y} = 0)| + |\mathcal{D}_0(\widehat{Y} = 1) - \mathcal{D}_1(\widehat{Y} = 1)|\right)$$
$$\leq |\gamma_0 - \gamma_1|$$
$$= |\mathcal{D}_0(Y = 0) - \mathcal{D}_1(Y = 0)|$$
$$= |\mathcal{D}_0(Y = 1) - \mathcal{D}_1(Y = 1)|$$
$$= \Delta_{\mathrm{BR}}(\mathcal{D}_0, \mathcal{D}_1) = \Delta_{\mathrm{DP}}(Y),$$

completing the proof. ∎

## A.5  PROOF OF THEOREM 3.2

**Theorem 3.2.** Assume the conditions in Proposition 3.1 hold and let $\widehat{Y} = h(g(X))$ be the classifier, then $\mathrm{Err}_{\mathcal{D}_0}(\widehat{Y}) + \mathrm{Err}_{\mathcal{D}_1}(\widehat{Y}) \leq 2\mathrm{BER}_{\mathcal{D}}(\widehat{Y} \,||\, Y)$.

*Proof.* First, by the law of total probability, we have:

$$\text{Err}_{\mathcal{D}_0}(\widehat{Y}) + \text{Err}_{\mathcal{D}_1}(\widehat{Y}) = \mathcal{D}_0(Y \neq \widehat{Y}) + \mathcal{D}_1(Y \neq \widehat{Y})$$
$$= \mathcal{D}_0^1(\widehat{Y} = 0)\mathcal{D}_0(Y = 1) + \mathcal{D}_0^0(\widehat{Y} = 1)\mathcal{D}_0(Y = 0) + \mathcal{D}_1^1(\widehat{Y} = 0)\mathcal{D}_1(Y = 1) + \mathcal{D}_1^0(\widehat{Y} = 1)\mathcal{D}_1(Y = 0)$$

Again, by Proposition 3.1, the classifier $\widehat{Y} = (h \circ g)(X)$ satisfies equalized odds, so we have $\mathcal{D}_0^1(\widehat{Y} = 0) = \mathcal{D}^1(\widehat{Y} = 0)$, $\mathcal{D}_0^0(\widehat{Y} = 1) = \mathcal{D}^0(\widehat{Y} = 1)$, $\mathcal{D}_1^1(\widehat{Y} = 0) = \mathcal{D}^1(\widehat{Y} = 0)$ and $\mathcal{D}_1^0(\widehat{Y} = 1) = \mathcal{D}^0(\widehat{Y} = 1)$:

$$= \mathcal{D}^1(\widehat{Y} = 0)\mathcal{D}_0(Y = 1) + \mathcal{D}^0(\widehat{Y} = 1)\mathcal{D}_0(Y = 0) + \mathcal{D}^1(\widehat{Y} = 0)\mathcal{D}_1(Y = 1) + \mathcal{D}^0(\widehat{Y} = 1)\mathcal{D}_1(Y = 0)$$
$$= \mathcal{D}^1(\widehat{Y} = 0) \cdot \big(\mathcal{D}_0(Y = 1) + \mathcal{D}_1(Y = 1)\big) + \mathcal{D}^0(\widehat{Y} = 1) \cdot \big(\mathcal{D}_0(Y = 0) + \mathcal{D}_1(Y = 0)\big)$$
$$\leq 2\mathcal{D}^1(\widehat{Y} = 0) + 2\mathcal{D}^0(\widehat{Y} = 1)$$
$$= 2\text{BER}_{\mathcal{D}}(\widehat{Y} \,\|\, Y),$$

which completes the proof. ∎

### A.6 PROOF OF THEOREM 3.3

**Theorem 3.3.** For any classifier $\widehat{Y}$, $\Delta_{\text{Err}}(\widehat{Y}) \leq \Delta_{\text{BR}}(\mathcal{D}_0, \mathcal{D}_1) \cdot \text{BER}_{\mathcal{D}}(\widehat{Y} \,\|\, Y) + 2\Delta_{\text{EO}}(\widehat{Y})$.

Before we give the proof of Theorem 3.3, we first prove the following two lemmas that will be used in the following proof.

**Lemma A.1.** Define $\gamma_a := \mathcal{D}_a(Y = 0), \forall a \in \{0, 1\}$, then $|\gamma_0 \mathcal{D}_0^0(\widehat{Y} = 1) - \gamma_1 \mathcal{D}_1^0(\widehat{Y} = 1)| \leq |\gamma_0 - \gamma_1| \cdot \mathcal{D}^0(\widehat{Y} = 1) + \gamma_0 \mathcal{D}^0(A = 1)\Delta_{\text{EO}}(\widehat{Y}) + \gamma_1 \mathcal{D}^0(A = 0)\Delta_{\text{EO}}(\widehat{Y})$.

*Proof.* In order to prove the upper bound in the lemma, it suffices if we could give the desired upper bound for the following term

$$\left| |\gamma_0 \mathcal{D}_0^0(\widehat{Y} = 1) - \gamma_1 \mathcal{D}_1^0(\widehat{Y} = 1)| - |(\gamma_0 - \gamma_1)\mathcal{D}^0(\widehat{Y} = 1)| \right|$$
$$\leq \left| \big(\gamma_0 \mathcal{D}_0^0(\widehat{Y} = 1) - \gamma_1 \mathcal{D}_1^0(\widehat{Y} = 1)\big) - (\gamma_0 - \gamma_1)\mathcal{D}^0(\widehat{Y} = 1) \right|$$
$$= \left| \gamma_0(\mathcal{D}_0^0(\widehat{Y} = 1) - \mathcal{D}^0(\widehat{Y} = 1)) - \gamma_1(\mathcal{D}_1^0(\widehat{Y} = 1) - \mathcal{D}^0(\widehat{Y} = 1)) \right|,$$

following which we will have:

$$|\gamma_0 \mathcal{D}_0^0(\widehat{Y} = 1) - \gamma_1 \mathcal{D}_1^0(\widehat{Y} = 1)| \leq |(\gamma_0 - \gamma_1)\mathcal{D}^0(\widehat{Y} = 1)|$$
$$+ \left| \gamma_0(\mathcal{D}_0^0(\widehat{Y} = 1) - \mathcal{D}^0(\widehat{Y} = 1)) - \gamma_1(\mathcal{D}_1^0(\widehat{Y} = 1) - \mathcal{D}^0(\widehat{Y} = 1)) \right|,$$

and an application of the Bayes formula could finish the proof. To do so, let us first simplify $\mathcal{D}_0^0(\widehat{Y} = 1) - \mathcal{D}^0(\widehat{Y} = 1)$. Applying the Bayes's formula, we know that:

$$\mathcal{D}_0^0(\widehat{Y} = 1) - \mathcal{D}^0(\widehat{Y} = 1) = \mathcal{D}_0^0(\widehat{Y} = 1) - \big(\mathcal{D}_0^0(\widehat{Y} = 1)\mathcal{D}^0(A = 0) + \mathcal{D}_1^0(\widehat{Y} = 1)\mathcal{D}^0(A = 1)\big)$$
$$= \big(\mathcal{D}_0^0(\widehat{Y} = 1) - \mathcal{D}_0^0(\widehat{Y} = 1)\mathcal{D}^0(A = 0)\big) - \mathcal{D}_1^0(\widehat{Y} = 1)\mathcal{D}^0(A = 1)$$
$$= \mathcal{D}^0(A = 1)\big(\mathcal{D}_0^0(\widehat{Y} = 1) - \mathcal{D}_1^0(\widehat{Y} = 1)\big).$$

Similarly, for the second term $\mathcal{D}_1^0(\widehat{Y} = 1) - \mathcal{D}^0(\widehat{Y} = 1)$, we can show that:

$$\mathcal{D}_1^0(\widehat{Y} = 1) - \mathcal{D}^0(\widehat{Y} = 1) = \mathcal{D}^0(A = 0)\big(\mathcal{D}_1^0(\widehat{Y} = 1) - \mathcal{D}_0^0(\widehat{Y} = 1)\big).$$

Plug these two identities into above, we can continue the analysis with

$$\left| \gamma_0(\mathcal{D}_0^0(\widehat{Y} = 1) - \mathcal{D}^0(\widehat{Y} = 1)) - \gamma_1(\mathcal{D}_1^0(\widehat{Y} = 1) - \mathcal{D}^0(\widehat{Y} = 1)) \right|$$
$$= \left| \gamma_0 \mathcal{D}^0(A = 1)(\mathcal{D}_0^0(\widehat{Y} = 1) - \mathcal{D}_1^0(\widehat{Y} = 1)) - \gamma_1 \mathcal{D}^0(A = 0)(\mathcal{D}_1^0(\widehat{Y} = 1) - \mathcal{D}_0^0(\widehat{Y} = 1)) \right|$$
$$\leq \left| \gamma_0 \mathcal{D}^0(A = 1)(\mathcal{D}_0^0(\widehat{Y} = 1) - \mathcal{D}_1^0(\widehat{Y} = 1)) \right| + \left| \gamma_1 \mathcal{D}^0(A = 0)(\mathcal{D}_1^0(\widehat{Y} = 1) - \mathcal{D}_0^0(\widehat{Y} = 1)) \right|$$
$$\leq \gamma_0 \mathcal{D}^0(A = 1)\Delta_{\text{EO}}(\widehat{Y}) + \gamma_1 \mathcal{D}^0(A = 0)\Delta_{\text{EO}}(\widehat{Y}).$$

The first inequality holds by triangular inequality and the second one holds by the definition of equalized odds gap. ∎

**Lemma A.2.** Define $\gamma_a := \mathcal{D}_a(Y = 0), \forall a \in \{0, 1\}$, then $\left|(1 - \gamma_0)\mathcal{D}_0^1(\widehat{Y} = 0) - (1 - \gamma_1)\mathcal{D}_1^1(\widehat{Y} = 0)\right| \leq |\gamma_0 - \gamma_1| \cdot \mathcal{D}^1(\widehat{Y} = 0) + (1 - \gamma_0)\mathcal{D}^1(A = 1)\Delta_{\mathrm{EO}}(\widehat{Y}) + (1 - \gamma_1)\mathcal{D}^1(A = 0)\Delta_{\mathrm{EO}}(\widehat{Y})$.

*Proof.* The proof of this lemma is symmetric to the previous one, so we omit it here. ∎

Now we are ready to prove Theorem 3.3:

*Proof of Theorem 3.3.* First, by the law of total probability, it is easy to verify that following identity holds for $a \in \{0, 1\}$:

$$\mathcal{D}_a(\widehat{Y} \neq Y) = \mathcal{D}_a(Y = 1, \widehat{Y} = 0) + \mathcal{D}_a(Y = 0, \widehat{Y} = 1)$$
$$= (1 - \gamma_a)\mathcal{D}_a^1(\widehat{Y} = 0) + \gamma_a\mathcal{D}_a^0(\widehat{Y} = 1).$$

Using this identity, to bound the error gap, we have:

$$|\mathcal{D}_0(Y \neq \widehat{Y}) - \mathcal{D}_1(Y \neq \widehat{Y})| = \left|((1 - \gamma_0)\mathcal{D}_0^1(\widehat{Y} = 0) + \gamma_0\mathcal{D}_0^0(\widehat{Y} = 1)) - ((1 - \gamma_1)\mathcal{D}_1^1(\widehat{Y} = 0) + \gamma_1\mathcal{D}_1^0(\widehat{Y} = 1))\right|$$
$$\leq \left|\gamma_0\mathcal{D}_0^0(\widehat{Y} = 1) - \gamma_1\mathcal{D}_1^0(\widehat{Y} = 1)\right| + \left|(1 - \gamma_0)\mathcal{D}_0^1(\widehat{Y} = 0) - (1 - \gamma_1)\mathcal{D}_1^1(\widehat{Y} = 0)\right|.$$

Invoke Lemma A.1 and Lemma A.2 to bound the above two terms:

$$|\mathcal{D}_0(Y \neq \widehat{Y}) - \mathcal{D}_1(Y \neq \widehat{Y})|$$
$$\leq \left|\gamma_0\mathcal{D}_0^0(\widehat{Y} = 1) - \gamma_1\mathcal{D}_1^0(\widehat{Y} = 1)\right| + \left|(1 - \gamma_0)\mathcal{D}_0^1(\widehat{Y} = 0) - (1 - \gamma_1)\mathcal{D}_1^1(\widehat{Y} = 0)\right|$$
$$\leq \gamma_0\mathcal{D}^0(A = 1)\Delta_{\mathrm{EO}}(\widehat{Y}) + \gamma_1\mathcal{D}^0(A = 0)\Delta_{\mathrm{EO}}(\widehat{Y})$$
$$+ (1 - \gamma_0)\mathcal{D}^1(A = 1)\Delta_{\mathrm{EO}}(\widehat{Y}) + (1 - \gamma_1)\mathcal{D}^1(A = 0)\Delta_{\mathrm{EO}}(\widehat{Y})$$
$$+ |\gamma_0 - \gamma_1|\mathcal{D}^0(\widehat{Y} = 1) + |\gamma_0 - \gamma_1|\mathcal{D}^1(\widehat{Y} = 0),$$

Realize that both $\gamma_0, \gamma_1 \in [0, 1]$, we have:

$$\leq \mathcal{D}^0(A = 1)\Delta_{\mathrm{EO}}(\widehat{Y}) + \mathcal{D}^0(A = 0)\Delta_{\mathrm{EO}}(\widehat{Y}) + \mathcal{D}^1(A = 1)\Delta_{\mathrm{EO}}(\widehat{Y}) + \mathcal{D}^1(A = 0)\Delta_{\mathrm{EO}}(\widehat{Y})$$
$$+ |\gamma_0 - \gamma_1|\mathcal{D}^0(\widehat{Y} = 1) + |\gamma_0 - \gamma_1|\mathcal{D}^1(\widehat{Y} = 0)$$
$$= 2\Delta_{\mathrm{EO}}(\widehat{Y}) + |\gamma_0 - \gamma_1|\mathcal{D}^0(\widehat{Y} = 1) + |\gamma_0 - \gamma_1|\mathcal{D}^1(\widehat{Y} = 0)$$
$$= 2\Delta_{\mathrm{EO}}(\widehat{Y}) + \Delta_{\mathrm{BR}}(\mathcal{D}_0, \mathcal{D}_1) \cdot \mathrm{BER}_{\mathcal{D}}(\widehat{Y} \,\|\, Y),$$

which completes the proof. ∎

We also provide the proof of Corollary 3.1:

**Corollary 3.1.** For any joint distribution $\mathcal{D}$ and classifier $\widehat{Y}$, if $\widehat{Y}$ satisfies equalized odds, then $\max\{\mathrm{Err}_{\mathcal{D}_0}(\widehat{Y}), \mathrm{Err}_{\mathcal{D}_1}(\widehat{Y})\} \leq \Delta_{\mathrm{BR}}(\mathcal{D}_0, \mathcal{D}_1) \cdot \mathrm{BER}_{\mathcal{D}}(\widehat{Y} \,\|\, Y)/2 + \mathrm{BER}_{\mathcal{D}}(\widehat{Y} \,\|\, Y)$.

*Proof.* We first invoke Theorem 3.3, if $\widehat{Y}$ satisfies equalized odds, then $\Delta_{\mathrm{EO}}(\widehat{Y}) = 0$, which implies:

$$\Delta_{\mathrm{Err}}(\widehat{Y}) = \left|\mathrm{Err}_{\mathcal{D}_0}(\widehat{Y}) - \mathrm{Err}_{\mathcal{D}_1}(\widehat{Y})\right| \leq \Delta_{\mathrm{BR}}(\mathcal{D}_0, \mathcal{D}_1) \cdot \mathrm{BER}_{\mathcal{D}}(\widehat{Y} \,\|\, Y).$$

On the other hand, by Theorem 3.2, we know that

$$\mathrm{Err}_{\mathcal{D}_0}(\widehat{Y}) + \mathrm{Err}_{\mathcal{D}_1}(\widehat{Y}) \leq 2\mathrm{BER}_{\mathcal{D}}(\widehat{Y} \,\|\, Y).$$

Combine the above two inequalities and recall that $\max\{a, b\} = (|a + b| + |a - b|)/2, \forall a, b \in \mathbb{R}$, yielding:

$$\max\{\mathrm{Err}_{\mathcal{D}_0}(\widehat{Y}), \mathrm{Err}_{\mathcal{D}_1}(\widehat{Y})\} = \frac{|\mathrm{Err}_{\mathcal{D}_0}(\widehat{Y}) - \mathrm{Err}_{\mathcal{D}_1}(\widehat{Y})| + |\mathrm{Err}_{\mathcal{D}_0}(\widehat{Y}) + \mathrm{Err}_{\mathcal{D}_1}(\widehat{Y})|}{2}$$
$$\leq \frac{\Delta_{\mathrm{BR}}(\mathcal{D}_0, \mathcal{D}_1) \cdot \mathrm{BER}_{\mathcal{D}}(\widehat{Y} \,\|\, Y) + 2\mathrm{BER}_{\mathcal{D}}(\widehat{Y} \,\|\, Y)}{2}$$
$$= \Delta_{\mathrm{BR}}(\mathcal{D}_0, \mathcal{D}_1) \cdot \mathrm{BER}_{\mathcal{D}}(\widehat{Y} \,\|\, Y)/2 + \mathrm{BER}_{\mathcal{D}}(\widehat{Y} \,\|\, Y),$$

completing the proof. ∎

## B  EXPERIMENTAL DETAILS

### B.1  THE ADULT EXPERIMENT

For the baseline network NODEBIAS, we implement a three-layer neural network with ReLU as the hidden activation function and logistic regression as the target output function. The input layer contains 114 units, and the hidden layer contains 60 hidden units. The output layer only contain one unit, whose output is interpreted as the probability of $\mathcal{D}(\widehat{Y} = 1 \mid X = x)$.

For the adversary in FAIR and LAFTR, again, we use a three-layer feed-forward network. Specifically, the input layer of the adversary is the hidden representations of the baseline network that contains 60 units. The hidden layer of the adversary network contains 50 units, with ReLU activation. Finally, the output of the adversary also contains one unit, representing the adversary's inference probability $\mathcal{D}(\widehat{A} = 1 \mid Z = z)$. The network structure of the adversaries in both CFAIR and CFAIR-EO are exactly the same as the one used in FAIR and LAFTR, except that there are two adversaries, one for $\mathcal{D}^0(\widehat{A} = 1 \mid Z = z)$ and one for $\mathcal{D}^1(\widehat{A} = 1 \mid Z = z)$.

The hyperparameters used in the experiment are listed in Table 2.

Table 2: Hyperparameters used in the Adult experiment.

| | |
|---|---|
| Optimization Algorithm | AdaDelta |
| Learning Rate | 1.0 |
| Batch Size | 512 |
| Training Epochs $\lambda \in \{0.1, 1.0, 10.0, 100.0, 1000.0\}$ | 100 |

### B.2  THE COMPAS EXPERIMENT

Again, for the baseline network NODEBIAS, we implement a three-layer neural network with ReLU as the hidden activation function and logistic regression as the target output function. The input layer contains 11 units, and the hidden layer contains 10 hidden units. The output layer only contain one unit, whose output is interpreted as the probability of $\mathcal{D}(\widehat{Y} = 1 \mid X = x)$.

For the adversary in FAIR and LAFTR, again, we use a three-layer feed-forward network. Specifically, the input layer of the adversary is the hidden representations of the baseline network that contains 60 units. The hidden layer of the adversary network contains 10 units, with ReLU activation. Finally, the output of the adversary also contains one unit, representing the adversary's inference probability $\mathcal{D}(\widehat{A} = 1 \mid Z = z)$. The network structure of the adversaries in both CFAIR and CFAIR-EO are exactly the same as the one used in FAIR and LAFTR, except that there are two adversaries, one for $\mathcal{D}^0(\widehat{A} = 1 \mid Z = z)$ and one for $\mathcal{D}^1(\widehat{A} = 1 \mid Z = z)$.

The hyperparameters used in the experiment are listed in Table 3.

Table 3: Hyperparameters used in the COMPAS experiment.

| | |
|---|---|
| Optimization Algorithm | AdaDelta |
| Learning Rate | 1.0 |
| Batch Size | 512 |
| Training Epochs $\lambda \in \{0.1, 1.0\}$ | 20 |
| Training Epochs $\lambda = 10.0$ | 15 |

