# OpenReview forum: "Conditional Learning of Fair Representations"
_ICLR.cc/2020/Conference — Accept (Spotlight)_

### Official Review · AnonReviewer2 · 2019-10-23
**Official Blind Review #2**

**Rating:** 6

**Review:**

This paper focuses on learning representations which can simultaneously achieve equalized odds and accuracy parity without impacting demographic parity. The authors show both theoretically and empirically that the proposed algorithm show better utility-fairness tradeoff on balanced datasets. This is indeed a useful result. Overall, I liked the presentation of the paper including motivation and background of other methods. Theory is sound, and experiments are sufficient.   Therefore, I do not have any major concern. Some of minor concerns are mentioned below:

1. I see some ambiguity in the definition of the classifiers. h is defined to be deterministic in section 2; whereas, later \hat{Y}, which I believe is treated at h(g(x)) for some h, is taken to be randomized. Also, the theorems are mentioned with respect to h making them specific to deterministic classifiers according to the current definition.

2. I am not sure why the last statement of the second last paragraph on page 2 is true. Can you please explain? There should be an additional condition on distribution D, which is not clear at that moment in the paper.

3. I am wondering if the utility can be maintained for imbalanced datasets by taking two parameters \lambda_1 and \lambda_2 for BER_{D^0} and BER_{D^1} in equation 2. Did the authors check when we have different regularization parameters for both terms? If yes, then what was the conclusion?

4. Please write theorems as fully independent statements. "Assume the conditions in Proposition 3.2" is probably not the right way to start a theorem. Other way is to mention the conditions separately and then use it throughout the paper.

5. Can you please write or elaborate the final optimization problem after section 3.4?

6. How did the authors construct the optimal classifiers in the experiments for real data? Can you provide some details?

Typo: "sensitive attribute A, then the second term" --> remove then

----  After Rebuttal ---

I thank the authors for providing response to my questions. At this point, I am going to keep the same score.

**Experience Assessment:**

I have read many papers in this area.

**Review Assessment: Checking Correctness Of Derivations And Theory:**

I assessed the sensibility of the derivations and theory.

**Review Assessment: Checking Correctness Of Experiments:**

I assessed the sensibility of the experiments.

**Review Assessment: Thoroughness In Paper Reading:**

I read the paper thoroughly.

---

> ### Author Response · Authors · 2019-11-07
> **Response**
>
> We thank the reviewer for the positive and encouraging comments of our work. We would like to answer your questions below.
>
> # Deterministic/Randomized Classifiers
> Yes, $\widehat{Y} = h(g(X))$, but the hypothesis $h(\cdot)$ defined in this work could be randomized, as long as the randomness is independent of $Y$. The only requirement here is that $h(\cdot)\in\{0, 1\}$. One way to think about this is that $h = h(R(g(X)), \epsilon)$, where $R(g(X))\in[0, 1]$ is a score function and $\epsilon\sim U(0, 1)$ is uniformly random variable, and $h(\cdot) = 1$ iff $\epsilon \leq R(g(X))$.
>
> # Last statement of the second last paragraph
> Note that here since $Y\in\{0, 1\}$, so we know $\mathcal{D}(Y)$ is a binary distribution. We use $\mathcal{D}(Y)$ to denote the marginal distribution of $Y$ from the joint distribution $\mathcal{D}$. The rest follows from the definition of total variation and $\Delta_{\text{BR}}(\cdot, \cdot)$.
>
> # Different Regularization parameters
> Thanks for the invaluable suggestion! We attributed the utility loss on imbalanced data to the first term of the objective function instead of the BER_{D^0} and BER_{D^1} terms. To see this, let's compare the blue curve and the green curve in the fourth plot of Figure 1. Note that the blue curve has less utility loss compared with the green one, and the only difference here is that the green curve corresponds to $BER_D(\widehat{Y} || Y)$ while the blue one corresponds to the usual cross-entropy loss. BER_{D^0} and BER_{D^1} terms are shared in both curves.
>
>
> Nevertheless, we performed additional experiments to valid the effects of different regularization parameters for BER_{D^0} and BER_{D^1}. Specifically, since the marginal distribution of Y = 1 roughly equals 0.25 in the Adult dataset, we use $0.5\lambda$ for BER_{D^0} and $1.5\lambda$ for BER_{D^1} with varying $\lambda$. The results are shown as follows:
>                                                     $\lambda = 0.1$     $\lambda = 1.0$     $\lambda = 10.0$    $\lambda = 100.0$   $\lambda = 1000.0$
> CFair, Err0 + Err1 ($0.5\lambda$, $1.5\lambda$):    0.3655441681        0.3687420002        0.4167540332        0.4677270783        0.5498600632
> CFair, Err0 + Err1 ($\lambda$, $\lambda$):          0.3641129377        0.3886518694        0.4138039334        0.4803962055        0.5368310545
> CFair-EO, Err0 + Err1 ($\lambda$, $\lambda$):       0.2954339377        0.287374556         0.2879574589        0.3122427609        0.3236335696
>
> From the above experimental results, we can confirm that even with different regularization parameters for BER_{D^0} and BER_{D^1}, the utility loss is roughly the same as the one with the same regularization parameters, and both of them are worse than CFair-EO (blue curve), which is mainly due to that CFair-EO uses the normal cross-entropy loss as target loss instead of the balanced error rate.
>
> # Proposition 3.2 and Optimization formulation
> Thanks for the suggestion, and we have updated the manuscript accordingly. The details about the final optimization objective function can now be found in Section 3.4.
>
> # Construction of the optimal classifier
> The details of the classifier and the optimization algorithm are described at the end of Section 4.1 and Appendix B. In a nutshell, we use simultaneous gradient update to implement the gradient descent-ascent algorithm to optimize the target classifier until convergence.

---

### Official Review · AnonReviewer1 · 2019-10-23
**Official Blind Review #1**

**Rating:** 6

**Review:**

Summary: Authors extend on work that attempts to learn fair data representation (features) and propose an algorithm (which is a modification of a loss essentially) and show that it allows to achieve accuracy and equalized odds parity, and show that while achieving equalized odds they don't hurt demographic parity. The experiments demonstrate utility of the algorithm for balanced datasets (without sacrificing performance to fairness)

Disclaimer: I am completely out of this area
But it is an easy read and an interesting angle. Authors show that simple modification of the loss makes it more fair (in a sense).  The experiments are somewhat thin.

Min max problem in Section 3 - I think it requires more intro for people outside. I assume you use an architecture that extracts the representation and has two head (outputs) - h and h'. For a modified loss, do you have 3 outputs - h, h' and h''? A pic with an architecture would be really helpful

Probably more datasets are required to have a more convincing empirical story

Section 3.4: Even though you can't directly optimize for BER, there are ways that can work, instead of just replacing it with CE, for example this https://arxiv.org/pdf/1608.04802.pdf

One critique is based on 3.4 I don't understand  how can this be extended to multiple axis of intersecting groups- e.g. not just mutually exclusive race values, but also gender for example.

**Experience Assessment:**

I do not know much about this area.

**Review Assessment: Checking Correctness Of Derivations And Theory:**

I assessed the sensibility of the derivations and theory.

**Review Assessment: Checking Correctness Of Experiments:**

I assessed the sensibility of the experiments.

**Review Assessment: Thoroughness In Paper Reading:**

I read the paper at least twice and used my best judgement in assessing the paper.

---

> ### Author Response · Authors · 2019-11-07
> **Response**
>
> We would like to thank the reviewer for providing the thoughtful comments. We attempt to answer your questions below.
>
> # Min-max problem
> Yes, our architecture essentially contains 4 parts, including one feature transformation component, one task classifier, and two adversarial classifiers, one for conditional distribution of $Y = 0$ and the other for $Y = 1$.
>
> # Experiments
> We carefully design our experiments to contain two datasets, one balanced and one imbalanced, to verify the performance of the proposed algorithm versus other approaches on these two datasets. Both the Adult and the COMPAS datesets are widely used in the literature for classification problems, which is the main focus of this work. We believe our current experiments are sufficient to demonstrate our theory (as commented by R2) and we anticipate similar experimental conclusions could be observed on other tasks as well. For example, both [1] and [2] consider and use these two datasets for empirical verifications of the corresponding methods proposed therein.
>
> # Balanced Error Rate
> Yes, we fully agree with the reviewer that there are many other convex relaxations of the intractable balanced error rate function besides the one used in our work, and we are happy to cite this work and have a discussion on it.
>
> # Multiple Groups
> The proposed algorithm could be easily extended to work under the setting where there are multiple sensitive attributes: we can simply define a new sensitive attribute that is the direct product of all the existing sensitive attributes. However, this will still lead to a sensitive attribute that contains exclusive memberships. A naive extension that also works for intersecting sensitive attributes is to add two more BER terms for each new sensitive attribute. However, this will inevitably lead to tradeoffs between multiple sensitive attributes, since they could be correlated by themselves.
>
> [1].    One-network Adversarial Fairness, Adel et al., AAAI 2019.
> [2].    A Reductions Approach to Fair Classification., FATML 2017.

---

### Official Review · AnonReviewer3 · 2019-10-29
**Official Blind Review #3**

**Rating:** 6

**Review:**

Update:

Thanks for providing additional results on the Adult dataset. I have increased my score. However I'd be nice to also see the balanced accuracy (i.e. sum of TPRs for each class divided by 2) results and compare to baseline trained with oversampling or re-weighted loss.

I would suggest authors to add more extensive comparisons to other methods using Adult dataset. There are quite a lot of papers in the fairness literature that experiment with the Adult dataset. They focus on different metrics and there is probably no method that is uniformly the best. Your paper demonstrated that your approach can succeed in achieving accuracy parity, but it would be good to also show tradeoffs with other metrics (in addition to DP). I was able to do some back-of-the-envelop calculations and your results seem fine, but a clear comparison would be good.

Below are some examples of the papers studying Adult dataset from the fairness angle:
[1] Mitigating unwanted biases with adversarial learning. Zhang et al., 2018. (already cited, but no comparison)
[2] What’s in a Name? Reducing Bias in Bios without Access to Protected Attributes. Romanov et al., 2019.
[3] Learning fair predictors with Sensitive Subspace Robustness. Yurochkin et al., 2019.


---------------------------------------------------------------------------------------------------------------------------------------------------------

This paper proposes an adversarial representation learning approach. The key difference with prior work is that the objective function is built around balanced error rates, one for classes, that is eventually used for classification, and two adversarial for predicting each of the protected attributes. Authors argue that proposed approach can simultaneously achieve accuracy parity and equalized odds.

The notion of accuracy parity does not seem to be very meaningful. For example, predicting uniformly at random seems like an intuitively fair classifier with EO gap 0 and DP gap 0. However it will not necessarily have error gap of 0 (i.e. satisfy accuracy parity), making me wonder if the notion of accuracy parity makes much sense.

I am not really sure what is the Err0 + Err1 metric used in Figures 1 and 2. Is it not normalized and can vary between 0 and 2? In which case it seems counterintuitive for performance quantification. If it is normalized, then results on COMPAS do not make sense. Err0 + Err1 of all methods is above 60%, which is worse than predicting uniformly at random.

Please report your TPRs for classes grouped by protected attribute when reporting the results in the context of group fairness. Further for Adult dataset, reporting balanced TPR as a measure of accuracy seems to make more sense given the class imbalance.



**Experience Assessment:**

I have published one or two papers in this area.

**Review Assessment: Checking Correctness Of Derivations And Theory:**

I assessed the sensibility of the derivations and theory.

**Review Assessment: Checking Correctness Of Experiments:**

I assessed the sensibility of the experiments.

**Review Assessment: Thoroughness In Paper Reading:**

I read the paper at least twice and used my best judgement in assessing the paper.

---

> ### Author Response · Authors · 2019-11-07
> **Response and Clarification about some misconceptions**
>
> We would like to thank the reviewer for providing the thoughtful comments, and we hope our following response could help to clarify some misconceptions in the review.
>
> # Accuracy parity:
> First, we would like to kindly point out that predicting uniformly at random WILL have error gap 0 no matter what the underlying distribution is. To see this, let $\widehat{Y}$ be a uniformly random predictor that outputs 0/1. Then for any distribution $\mathcal{D}$, we have $\text{Err}(\widehat{Y}) = \Pr_{\mathcal{D}}(Y\neq Y) = \Pr_\mathcal{D}(Y=1,\widehat{Y} = 0) + \Pr_\mathcal{D}(Y=0,\widehat{Y} = 1) = \Pr_\mathcal{D}(Y=1)\cdot\Pr(\widehat{Y} = 0) + \Pr_\mathcal{D}(Y=0)\cdot\Pr(\widehat{Y} = 1) = \frac{1}{2}(\Pr_\mathcal{D}(Y=1) + \Pr_\mathcal{D}(Y=0)) = 1/2$. This implies that the errors of $\widehat{Y}$ on both demographic subgroups will be 1/2, leading to accuracy parity by definition.
>
> Furthermore, the notion of accuracy parity has been widely discussed and used in the literature as a principle to design fair algorithms. To the best of our knowledge, this dates at least back to 2015 by Zafar et al. [1], and has received increasing attention since then [2-4, 7]. See [6, Table 1] for a thorough investigation on this topic. In fact, as demonstrated by researchers from MIT Media Lab and Microsoft Research [5], three widely used commercial face recognition systems exhibit substantial accuracy disparities between four groups: darker females, lighter females, darker males, and lighter males. This result has already brought huge public attention [8-10] and calls for commercial face recognition systems that (at least approximately) satisfy accuracy parity.
>
> Last but not least, from a theoretical point of view, it is well-known that any two of the three fairness definitions (DP, EO, Predictive Rate Parity) cannot hold simultaneously except in degenerate cases [11]. Nevertheless, accuracy parity is free of such inherent incompatibility, and we believe it's both interesting (from a theoretical standpoint) and urgent (from a practical viewpoint) to have algorithms that can achieve both accuracy parity and equalized odds.
>
> [1].    Fairness Constraints: Mechanisms for Fair Classification, Zafar et al., 2015.
> [2].    Fairness Beyond Disparate Treatment & Disparate Impact: Learning Classification without Disparate Mistreatment, Zafar et al., 2016.
> [3].    Demonstrating accuracy equity and predictive parity performance of the compas risk scales in broward county. Dieterich et al., 2016.
> [4].    Fairness in Criminal Justice Risk Assessments: The State of the Art, Berk et al., 2017.
> [5].    Gender Shades: Intersectional Accuracy Disparities in Commercial Gender Classification, Buolamwini et al., 2018.
> [6].    Fairness Definitions Explained, Verma et al., 2018.
> [7].    Inherent Tradeoffs in Learning Fair Representations, Zhao et al., 2019.
> [8].    The Verge blog: The tech industry doesn’t have a plan for dealing with bias in facial recognition (https://www.theverge.com/2018/7/26/17616290/facial-recognition-ai-bias-benchmark-test).
> [9].    Insurance Journal: MIT Researcher Exposing Bias in Facial Recognition Tech Triggers Amazon’s Wrath (https://www.insurancejournal.com/news/national/2019/04/08/523153.htm).
> [10].   The New York Times: Facial Recognition Is Accurate, if You’re a White Guy (https://www.nytimes.com/2018/02/09/technology/facial-recognition-race-artificial-intelligence.html).
> [11].   Fairness in Machine Learning, NIPS 2017 Tutorial.
>
>
> # Err0 + Err1 in Figure 1 and 2:
> Both Err0 and Err1 range between 0 and 1, so the sum of them range between 0 and 2. The metric Err0 + Err1 shown in Figure 1 and 2 is not normalized, but all the curves in the last plot share this same metric, so we believe they still provide a fair comparison among these different algorithms. The main reason we use this metric is that we would like it to be consistent with the theoretical result presented in Theorem 2.1, which precisely gives a lower bound in terms of Err0 + Err1.
>
>
> # True Positive Rate
> True positive rate parity corresponds to a sub-case of equalized odds known as equality of opportunity, which essentially reduces to our definition $\Delta_{\text{EO}}(\widehat{Y})$ when only $y = 1$ is considered (see Definition 3.3). Hence it is clear that our metric of $\Delta_{\text{EO}}(\widehat{Y})$ is actually more stringent than TPR parity since it's an upper bound of the TPR gap. The results on this could be found in the second plot of Figure 1 and 2.
>
>
> We hope our response has answered the reviewer's questions and helped further clarify the description and contribution of our work.

---

> > ### Comment · AnonReviewer3 · 2019-11-14
> > **Thanks for the rebuttal! I'd still like to see TPR results on Adult as I requested (not TPR gaps).**
> >
> > Thank you for the rebuttal!
> >
> > Of course you are right about the 0 error gap for a classifier predicting uniformly at random and I am sorry for a bad example. However, if we consider another "fair" classifier always predicting 1, then error gap is 0 only if probability of 1 is same across protected groups, which is often not the case. I also briefly went through some references you suggested, in particular [6] cites [3] for "Overall accuracy equality" and [3] in turn says "Overall accuracy equality is not commonly used because it does not distinguish between accuracy for successes and accuracy for failures". [1] seems to focus on disparate impact, i.e. eq. (1) in their paper (which would categorize an "always 1" classifier as fair) - could you please clarify where they study accuracy parity?
> >
> > While I still remain unconvinced that accuracy parity is preferable to other fairness notions, I acknowledge that this paper makes a contribution in the direction of improving fairness in ML. I'd be willing to increase my score if you could provide the TPRs by gender (not gaps) and overall balanced accuracy for the Adult dataset. The reason I am asking for these results is that TPR for females on Adult dataset is typically quite low when using a classifier without fairness considerations, however some results reported in the literature appear to "fix" the problem by essentially making TPR for males significantly lower. Accuracy at the same time does not seem to decrease by a lot because the overall proportion of the ">50k" class is small. I'd like to know if this is also the case for your method.

---

> > > ### Author Response · Authors · 2019-11-14
> > > **Response and Additional Experimental Results**
> > >
> > > We thank the reviewer for the time reading our response.
> > >
> > > # Constant predictor
> > > Yes, we agree with the reviewer that predictors that always output constant will not necessarily satisfy the accuracy parity, and they satisfy accuracy parity iff the base rates are equal. However, constant predictors are considered degenerate cases in binary classifications and they are not very useful.
> > >
> > > # Accuracy parity in practice
> > > Yes, [6] cites [3] as evidence of the use of accuracy parity in practice, and this is indeed the case. [3] mainly discusses the properties of famous COMPAS tool in criminal judgement, and it has stated several times in the manuscript that "the risk scales exhibit accuracy equity" (page 3), "The PP authors ignore evidence of accuracy equity and predictive
> > > parity of the COMPAS risk scales" (page 6), just to name a few.
> > >
> > > Please note that Eq. (1) in [1] is the related work in the literature cited by the authors of [1], while the approach proposed by [1] is in Section 3.3 "Maximizing Fairness Under Accuracy Constraints". Please further check Eq. (7) and Eq. (8) in [1], where the constraint in Eq. (7) basically corresponds to a multiplicative approximation of the accuracy parity constraint.
> > >
> > > # TPRs
> > > As requested by the reviewer, here we list the experimental results on the Adult dataset in terms of TPR by group, balanced TPR as well as the overall TPR, followed by our analysis of the results.
> > >
> > >                     NoDebias
> > > TPR | A = 0         0.6248806872
> > > TPR | A = 1         0.5278276481
> > > Balanced TPR        0.5763541677
> > > TPR                 0.6102702703
> > >
> > >                     CFair-EO, $\lambda = 0.1$       CFair-EO, $\lambda = 1.0$       CFair-EO, $\lambda = 10.0$       CFair-EO, $\lambda = 100.0$        CFair-EO, $\lambda = 1000.0$
> > > TPR | A = 0         0.6598790964	                0.6350620426	                0.6172446707	                 0.5841552657	                    0.4607063315
> > > TPR | A = 1         0.5709156194	                0.6535008977	                0.6391382406	                 0.5709156194	                    0.4631956912
> > > Balanced TPR        0.6153973579	                0.6442814702	                0.6281914556	                 0.5775354425	                    0.4619510114
> > > TPR                 0.6464864865	                0.6378378378	                0.6205405405	                 0.5821621622	                    0.4610810811
> > >
> > >                     CFair, $\lambda = 0.1$       CFair, $\lambda = 1.0$       CFair, $\lambda = 10.0$       CFair, $\lambda = 100.0$        CFair, $\lambda = 1000.0$
> > > TPR | A = 0         0.8361438116	             0.8129175947	              0.8052815781	                0.7995545657	                0.7410117722
> > > TPR | A = 1         0.7091561939	             0.7935368043	              0.8330341113	                0.8671454219	                0.6606822262
> > > Balanced TPR        0.7726500028	             0.8032271995	              0.8191578447	                0.8333499938	                0.7008469992
> > > TPR                 0.817027027	                 0.81	                      0.8094594595	                0.8097297297	                0.7289189189
> > >
> > > Analysis:
> > > First, note that the only difference between CFair-EO and CFair is that CFair uses conditional (conditioned on $Y$) cross-entropy as the objective function while CFair-EO uses the the original global cross-entropy as the objective function.
> > >
> > > Next, since both CFair-EO and CFair align the conditional distributions of features (conditioned on $A$), we can see that the TPR gap decreases as the hyperparameter $\lambda$ increases. On the other hand, since larger $\lambda$ corresponds to focusing
> > > more on aligning the features, this causes the overall TPR to gradually go down, which appear in both CFair and CFair-EO.
> > >
> > > Now if we compare CFair-EO and CFair with the baseline NoDebias, we see that both the TPR | A = 0 and TPR | A = 1 of CFair are much higher than that of the baseline, whereas these two metrics are roughly the same in
> > > CFair-EO. Why this happens? This is precisely because we use the balanced error rate (BER) as our objective function in CFair. As we can see from Table 1, the Adult data has very skewed distribution in terms of the target label $Y$. In this case,
> > > the balanced error rate will help to put equal footing on errors from instances with labels $Y = 1$ and $Y = 0$. On the other hand, since both NoDebias and CFair-EO use the original global cross-entropy loss as the objective function, and the fact that
> > > the marginal distribution of $Y = 1$ is only around $10\%$ in the dataset, the TPR for these two methods are much lower.
> > >
> > > Again, we thank the reviewer's time and efforts for the comment and we hope our response and new experimental results help address your concerns.

---

### Author Response · Authors · 2019-11-07
**General Response**

We thank all the reviewers for the thoughtful comments and we answer each reviewer’s questions individually below. We also updated the draft based on reviewers' comments.

---

### Decision · Program_Chairs · 2019-12-19

**Decision:**

Accept (Spotlight)

**Comment:**

This paper provides a new algorithm for learning fair representation for two different fairness criteria--accuracy parity and equalized odds. The reviewers agree that the paper provides novel techniques, although the experiments may appear to be a bit weak. Overall, this paper gives new contributions to the fair representation learning literature.

The authors should consider citing and discussing the relationship with the following work:
A Reductions Approach to Fair Classification., ICML 2018